# Hybrid achromatic microlenses with high numerical apertures and focusing efficiencies across the visible

Corey A. Richards[1,2,3], Christian R. Ocier[1,2,3], Dajie Xie[1,2,3], Haibo Gao[1,2,3], Taylor Robertson[4], Lynford L. Goddard [3,5,6], Rasmus E. Christiansen [7], David G. Cahill [1,2,8] & Paul V. Braun [1,2,3,8] ✉

Compact visible wavelength achromats are essential for miniaturized and lightweight optics. However, fabrication of such achromats has proved to be exceptionally challenging. Here, using subsurface 3D printing inside mesoporous hosts we densely integrate aligned refractive and diffractive elements, forming thin high performance hybrid achromatic imaging micro-optics. Focusing efficiencies of 51–70% are achieved for 15 μm thick, 90 μm diameter, 0.3 numerical aperture microlenses. Chromatic focal length errors of less than 3% allow these microlenses to form high-quality images under broadband illumination (400–700 nm). Numerical apertures upwards of 0.47 are also achieved at the cost of some focusing efficiency, demonstrating the flexibility of this approach. Furthermore, larger area images are reconstructed from an array of hybrid achromatic microlenses, laying the groundwork for achromatic light-field imagers and displays. The presented approach precisely combines optical components within 3D space to achieve thin lens systems with high focusing efficiencies, high numerical apertures, and low chromatic focusing errors, providing a pathway towards achromatic micro-optical systems.

Conventional refractive achromatic optics are ubiquitous, but their large volumes make them incompatible with many emerging technological platforms. Ultrathin, subwavelength diffractive optics such as metasurface lenses (metalenses) strive to greatly reduce the volume of achromatic imagers by replacing multiple refractive elements with a single flat surface[1]. However, metalenses experience significant trade-offs between achromaticity, signal-to-noise ratio, numerical aperture, and aperture size[2–4]. As a result, achromatic metalenses have been limited to near-infrared or infrared wavelengths[3, 5–9], small diameters (typically in the range of 10–40 μm)[4–6,8–13], low numerical apertures[3,5–7,9–11,13] (NAs) (<0.3), or reduced focusing efficiencies[7,11,14,15].

Overcoming this tradeoff would require the fabrication of nanopillars that are much taller, have a higher aspect ratio, and are more geometrically complex than what has been reported using visibly transparent materials[2–5]. A variety of devices, including ultracompact visible wavelength cameras[16–18], wide field of view compact near-eye displays and eyepieces[14,15,19–21], portable microscopes[22–25], high-resolution achromatic microlens array imagers[26–30], and charge-coupled devices (CCDs) with improved light collection[28,31,32] are therefore challenging to realize using metalenses.

Hybrid achromatic imaging systems overcome these limitations by combining diffractive and refractive optics, utilizing their

[1]Department of Materials Science and Engineering, University of Illinois Urbana-Champaign, Urbana, IL, USA. [2]Materials Research Laboratory, University of Illinois Urbana-Champaign, Urbana, IL, USA. [3]Beckman Institute for Advanced Science and Technology, University of Illinois Urbana-Champaign, Urbana, IL, USA. [4]Ansys Inc, Vancouver, BC, Canada. [5]Department of Electrical and Computer Engineering, University of Illinois Urbana-Champaign, Urbana, IL, USA. [6]Holonyak Micro and Nanotechnology Laboratory, University of Illinois Urbana-Champaign, Urbana, IL, USA. [7]Department of Civil and Mechanical Engineering, Technical University of Denmark, Kongens Lyngby, Denmark. [8]Department of Mechanical Science and Engineering, University of Illinois Urbana-Champaign, Urbana, IL, USA. ✉e-mail: pbraun@illinois.edu

complementary negative and positive dispersions to correct chromatic focusing errors[33,34]. This approach achieves an attractive balance between lens volume and achromaticity, with hybrid imagers realizing better chromatic correction than diffractive lenses and thinner geometries than conventional refractive doublets[33]. However, current hybrid optical systems require complex assembly with micron-level precision and external mechanical supports to align and integrate diffractive and refractive components[35–37]. Furthermore, producing large-area achromatic hybrid lens arrays is outside the purview of what can be achieved with current fabrication technologies. A method to seamlessly produce, integrate, and align diffractive and refractive elements with submicron accuracy and without the need for external supports would enable the advantages of hybrid systems while also minimizing volume, increasing ease of fabrication, and providing high-efficiency achromatic focusing.

Our newly developed fabrication process, coined Subsurface Controllable Refractive Index via Beam Exposure (SCRIBE), provides a path towards densely integrated hybrid optical systems. Via SCRIBE, polymeric structures are 3D printed within the pores of a silicon dioxide (PSiO$_2$) host medium substrate with submicron resolution using an ultrafast laser[38,39]. The host medium mechanically supports optical components in 3D space, enabling vertical integration with submicron resolution and nanoscale spatial precision. Optics fabricated within the pore volume of PSiO$_2$ experience low absorption and scattering losses due to the host's high void volume (>80% air), ultra-low refractive index ($n \sim 1.1$), and subwavelength pore diameter (<50 nm diameter)[40–42]. Furthermore, the local optical properties of printed structures can be tuned by modulating the laser exposure during printing. Our previously demonstrated subsurface micro-optical elements include gradient refractive index (GRIN) lenses and dispersion-engineered refractive achromatic doublets[38,43], highlighting the flexibility in SCRIBE's ability to form various types of optics within a tiny volume. However, despite SCRIBE's ability to render refractive doublets, the diameters of these doublet lenses cannot be easily scaled up without a significant increase in thickness. Therefore, our previously demonstrated optics are insufficient for generating highly compact visible wavelength achromats with high NAs, large diameters, and high focusing efficiencies.

Here, we leverage SCRIBE to create high NA (0.3) compact achromatic hybrid microlenses with diameters of ~90 μm, maximum focusing efficiencies of 51–70%, and a focal length error below 3% across the wavelength range of 488 to 633 nm. Our hybrid lenses comprise vertically integrated diffractive and refractive elements fabricated within the volume of a PSiO$_2$ host. Hybrid lenses with both plano-convex and GRIN refractive components are demonstrated with this approach to provide efficient achromatic focusing. A micron-sized gap is uniformly maintained between the diffractive and refractive lenses across the entire aperture. The refractive and diffractive optics are naturally aligned during fabrication, enabling the structure to form images under broadband white light illumination while minimizing volume and thickness.

As additional demonstrations, we fabricate higher NA hybrid microlenses by individually tailoring the geometries of the refractive and diffractive elements within the doublet without increasing the overall thickness (~15 μm thickness). An NA of 0.47 is achieved for white light imaging with focusing efficiencies >30% across the visible, a feat that is challenging for metalenses without experiencing major reductions in aperture size, achromaticity, or focusing efficiency[2–4]. Finally, we fabricate achromatic hybrid microlenses into an array to capture light-field information from a resolution target, which is a significant challenge using conventional polymer microlenses and is an important first step for full-color light-field imaging. The images formed by each individual microlens in the array are stitched together to reconstruct a larger area of white light image. This study presents a path to forming compact imagers that overcomes many limitations of

metalenses and traditional polymer micro-optics by using 3D printing to combine multiple optical elements within the pore volume of a host medium, achieving high, controllable NAs, high focusing efficiencies, good dispersion control, and large diameters while retaining a thickness of only 15 μm.

## Results
### Subsurface achromatic hybrid lenses
Compact, embedded hybrid doublets are fabricated using high-resolution multiphoton 3D patterning within a mesoporous host. A liquid resist (IP-Dip, n ~ 1.55) is soaked into the void volume of PSiO$_2$ and is polymerized via an ultrafast laser, locally replacing the air within the pores. A complete fabrication process flow can be found in Supplementary Information (SI) Section 1. As depicted in Fig. 1a, the porous host supports the printed optics in 3D space, allowing dense stacking and natural alignment of optical components. In the illustration, vertically integrated refractive and diffractive focusing elements work in concert to focus broadband white light to a single point. This concept forms the basis for low volume achromatic micro-optics with high focusing performance at visible wavelengths.

The geometries of the subsurface printed structures are experimentally imaged with confocal fluorescence microscopy (Fig. 1b–e), revealing the compact 3D objects inside the PSiO$_2$ host[44]. Hybrid lenses explored in this work include those with both geometric (Fig. 1b) and GRIN (Fig. 1c) refractive components positioned above diffractive lenses with a uniform 2 μm gap between each element. Dispersive diffractive lenses were also fabricated individually to serve as control samples (confocal fluorescence images in Fig. 1d, e) for comparing achromaticity, focusing efficiency, and imaging quality. The lower NA diffractive lens in Fig. 1d is identical to the one used for the hybrid lenses, whereas the higher NA diffractive lens in Fig. 1e was designed to have the same focal length as the hybrid lenses for a more direct comparison of focusing performance.

Hybrid lenses were designed with a target NA of 0.3 and aperture diameter of ~90 μm (a larger diameter is possible but is currently limited by our fabrication equipment, see Methods Section), values largely inaccessible to high-efficiency achromatic metalenses in the literature. There are any number of combinations of diffractive and refractive focusing powers that can achieve a given NA[33], as given by Eq. 1:

$$\Phi_{hyb}(\lambda) = \Phi_{ref}(\lambda) + \Phi_{dif}(\lambda) \tag{1}$$

where $\Phi_{hyb}(\lambda)$, $\Phi_{ref}(\lambda)$, and $\Phi_{dif}(\lambda)$ are the wavelength-dependent focusing powers of the hybrid lens, the refractive component, and the diffractive component, respectively. The focusing powers of the individual refractive and diffractive components are defined by Eqs. 2 and 3:

$$\Phi_{ref}(\lambda) = \frac{1}{f_{ref}(\lambda)} = \left(\frac{1}{R_f} - \frac{1}{R_b}\right) \cdot [n(\lambda) - 1] \tag{2}$$

$$\Phi_{dif}(\lambda) = \frac{1}{f_{dif}(\lambda)} = \frac{8M\lambda}{D^2} \tag{3}$$

in which $f_{ref}(\lambda)$ and $f_{dif}(\lambda)$ are the wavelength-dependent focal lengths, $R_f$ and $R_b$ are the radii of the refractive subcomponent's front and back surfaces, $n(\lambda)$ is the wavelength-dependent refractive index, M is the number of zones in the diffractive component, and D is the aperture diameter.

Each combination of $\Phi_{ref}(\lambda)$ and $\Phi_{dif}(\lambda)$ forms a hybrid lens with a different effective dispersion and a different total thickness. For our materials system, which employs moderate to low dispersion materials like silica and IP-Dip photoresin (see Methods Section), designs that

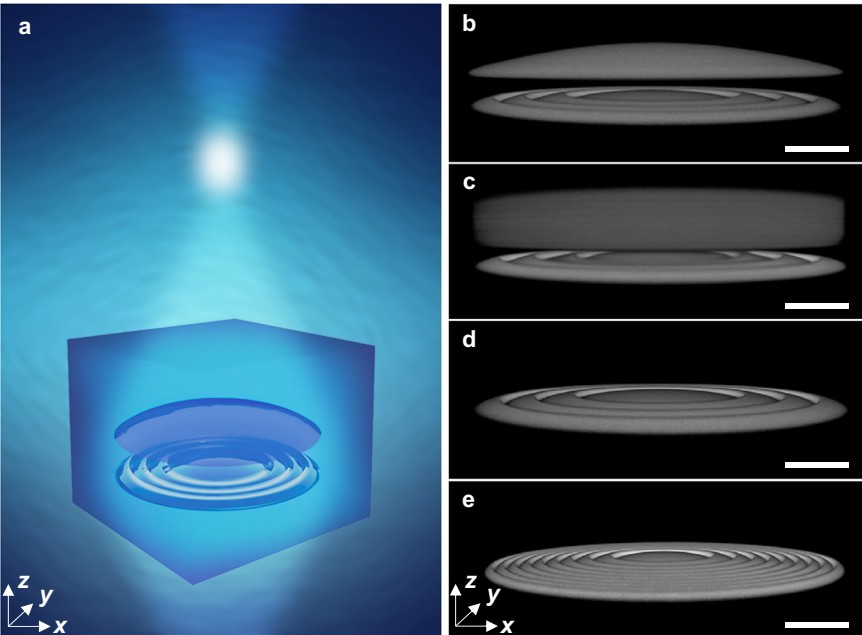

**Fig. 1 | Volumetric integration of diffractive and refractive components for compact hybrid achromatic microlenses. a** Artistic rendering of a subsurface hybrid lens focusing white light. The low refractive index PSiO$_2$ host medium is represented by a blue cube surrounding the printed diffractive and refractive optical components. In the real system, the cube extends in X and Y, but was truncated in the illustration. White light enters the hybrid lens' aperture from the bottom and is focused to a single point. **b**, **c** Geometric and GRIN hybrid doublet achromats. Experimentally measured confocal fluorescence images of achromatic hybrid doublets fabricated within the volume of the host medium. Both geometric (**b**) and GRIN (**c**) refractive components are integrated above flat diffractive lenses. **d**, **e** Lower and higher NA diffractive lenses. Experimentally measured confocal fluorescence images of diffractive lenses fabricated within the volume of the PSiO$_2$ host medium. The diffractive lenses are used as control samples. The diffractive lens shown in **d** is identical to the diffractive component used in the hybrid lenses. The diffractive lens in **e** has a higher NA and was designed to have the same focal length as the hybrid lenses at a wavelength of 633 nm, allowing for a more direct comparison. The black regions surrounding the experimentally imaged microlenses in **a**–**d** are occupied by PSiO$_2$. Scale bars are 15 μm.

utilize higher relative refractive focusing powers are generally able to better cancel out the strong negative dispersion of the diffractive lens[33]. However, according to Eq. 2, solutions with higher focusing power contributions from the refractive component require smaller radii of curvatures and, therefore, must be thicker for a given aperture diameter, which is undesirable in terms of bulk reduction and printing speed and accuracy. This tradeoff between volume and achromaticity is especially critical for hybrid systems with high NA. Therefore, for this initial work, refractive and diffractive components were selected to achieve a balance between achromaticity and overall thickness.

The hybrid lens in Fig. 1b was designed to have a maximum thickness of 15 μm and a focal length error below 3%, allowing the hybrid doublet to assume a compact structure while forming high-quality images. The design goals were experimentally met through the integration of a diffractive component with $\Phi_{dif}(633\,nm) = 0.0027\,\mu m^{-1}$, which translated to an NA of 0.118 at 633 nm (Fig. 1d), and a plano-convex aspheric refractive component with 11 μm thickness. The diffractive lens was fabricated with a thickness of 1.7 μm and the gap between the two subcomponents was 2 μm, resulting in an overall thickness of 14.7 μm.

The radial dependence of the thickness T(r) of the diffractive lens was designed by wrapping the phase function of a parabolic lens into zones that each accommodate a 2π phase shift according to the equations:

$$T(r) = T_{opt} \cdot \left( \frac{\frac{2\pi}{\lambda} \cdot \left( f_{dif}(\lambda) - \left( f_{dif}^2(\lambda) + r^2 \right)^{\frac{1}{2}} \right) + 2m\pi}{2\pi} + 1 \right) \quad (4)$$

$$r_m = \left( 2m\lambda f_{dif}(\lambda) + (m\lambda)^2 \right)^{\frac{1}{2}} \quad (5)$$

where T$_{opt}$ is the optimal thickness of the diffractive lens (T$_{opt}$ = 1.7 μm, which is close to the predicted value of $\frac{\lambda}{\Delta n} = \frac{0.633\mu m}{0.38} = 1.67\mu m$, SI Section 2 describes how T$_{opt}$ was experimentally determined), λ is the design wavelength, f$_{dif}$(λ) is the target focal length of the diffractive lens, m is the zone number (m = 0, 1, 2, 3, etc.), and r$_m$ is the radius of the mth zone[45]. A second hybrid doublet was also designed by replacing the plano-convex refractive lens with a cylindrical GRIN lens (Fig. 1c). This GRIN element exhibits an aspheric refractive index profile, a center-to-edge index contrast of roughly 0.35 (see SI Section 3), and a thickness of 13 μm. The overall thickness of this hybrid lens was slightly higher than the 15 μm target, but a nearly equivalent $\Phi_{hyb}$(λ) was maintained. The GRIN profile was produced by radially varying the local laser dosage during printing. See the Methods Section for a more detailed description of the fabrication parameters of each element.

## Focal length and focusing efficiency

The achromaticity of the diffractive and hybrid lenses was characterized at wavelengths of 488 nm (blue), 542 nm (green), 612 nm (orange), and 633 nm (red) using free-space lasers for illumination. A confocal microscope was used to capture X-Z focal profiles of one of each type of lens at each wavelength (X-Y focal plane scans available in SI Section 4). Figure 2 shows the measured X-Z focal profiles of a lower NA diffractive lens (Fig. 2a), a higher NA diffractive lens (Fig. 2b), a GRIN hybrid lens (Fig. 2c), and a geometric hybrid lens (Fig. 2d). The higher NA diffractive lens, the GRIN hybrid lens, and the geometric hybrid lens have similar focal lengths at 633 nm, but the hybrid lenses experienced a reduced chromatic focal length error across the visible compared to the diffractive lenses.

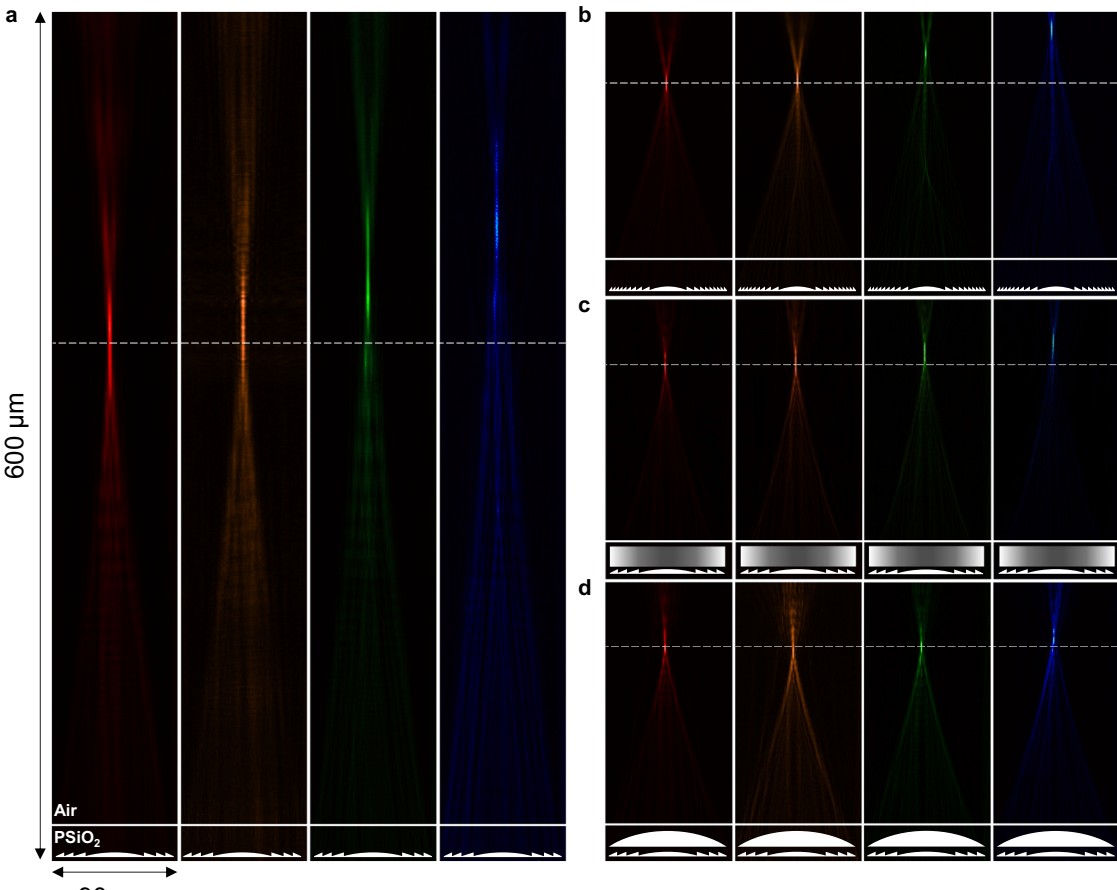

**Fig. 2 | Focal profiles of diffractive and hybrid lenses at visible wavelengths.**
**a**, **b** X-Z confocal scans of a lower NA diffractive lens (**a**) and a higher NA diffractive lens (**b**) focusing 633, 612, 542, and 488 nm light. These lenses experience strong negative dispersion. **c**, **d** X-Z confocal scans of a GRIN hybrid lens (**c**) and a geometric hybrid lens (**d**) focusing 633, 612, 542, and 488 nm light. These lenses experience reduced negative dispersion. For all lenses, the horizontal white dotted lines are the approximate reference focal planes for 633 nm light. The horizontal white solid lines represent the approximate interface between the PSiO₂ host medium and air. The lens cartoons represent the approximate locations of the printed lenses. All scans are 90 µm in X and are to scale with each other in Z.

To quantify fabrication errors between prints (which can be caused by printing laser fluctuations, local variations in the properties of PSiO₂ within the refractive component, or batch-to-batch variations in the properties of PSiO₂)[39, 46], ten separate lenses of each type were fabricated, and their focal lengths were averaged. The focal length measurement procedure is described in the Methods Section. Measured focal lengths are plotted in Fig. 3a, with each point representing the mean value of the ten printed lenses. The chromatic focal length errors of the geometric and GRIN hybrid lenses both averaged roughly 4 µm around the central focal length[3,8] (approximately 2.8% focal length error), where 542 nm is taken to be the central wavelength. The largest error occurs between 488 and 542 nm light due to the over-correction by the diffractive lens. The effective Abbe numbers of the printed hybrid doublets averaged around −20.3 (approximately six times less dispersive than a standard diffractive lens, which has an Abbe number of −3.45), as given by Eq. 6:

$$V_{hyb} = \frac{f_{short}f_{long}}{f_{center}} \cdot \frac{1}{f_{long} - f_{short}} = \frac{f(488nm)f(633nm)}{f(542nm)} \cdot \frac{1}{f(633nm) - f(488nm)} \quad (6)$$

where $V_{hyb}$ is the effective Abbe number of the hybrid element, and f(488 nm), f(542 nm), and f(633 nm) are the focal lengths of the hybrid lens at each wavelength. In comparison, the average focal length errors around the central wavelength were 55 µm (13% error) for the lower NA diffractive lenses and 20 µm (12% error) for the higher NA diffractive

lenses. The focal lengths were converted to NA at each wavelength, as shown in SI Section 5.

The hybrid lenses' focusing efficiencies were also characterized at each wavelength. The upper bound of focusing efficiencies achievable with our designs were estimated by broadband (488–633 nm) 3D finite-difference time-domain (FDTD) simulations using the software Ansys Lumerical. The simulated focusing efficiency was calculated by normalizing the intensity across the entire focal plane, integrating the intensity within the focal spot (radius of approximately three times the full width at half maximum (FWHM)), and dividing that value by the total intensity at the focal plane. Focusing efficiencies of around 75–90% were simulated across the visible for the diffractive lenses and hybrid achromats (SI Section 6). Simulation details are available in the Methods Section.

The focusing efficiencies of fabricated lenses were measured and plotted in Fig. 3b, with each point representing the average measured efficiency value across the 10 samples. Error bars indicate the minimum and maximum measured efficiencies, with variations occurring from random fluctuations in the printing laser and small local differences in the host medium's optical properties[39,46]. The measured focusing efficiency of the low NA diffractive lens ranges from roughly 70–80% across the visible, with the highest efficiency near the 633 nm design wavelength (purple squares). Geometric (red circles) and GRIN (blue triangles) hybrid doublets retained average focusing efficiencies of 45–64% (maximum focusing efficiencies 51–70%) and 35–62% (maximum

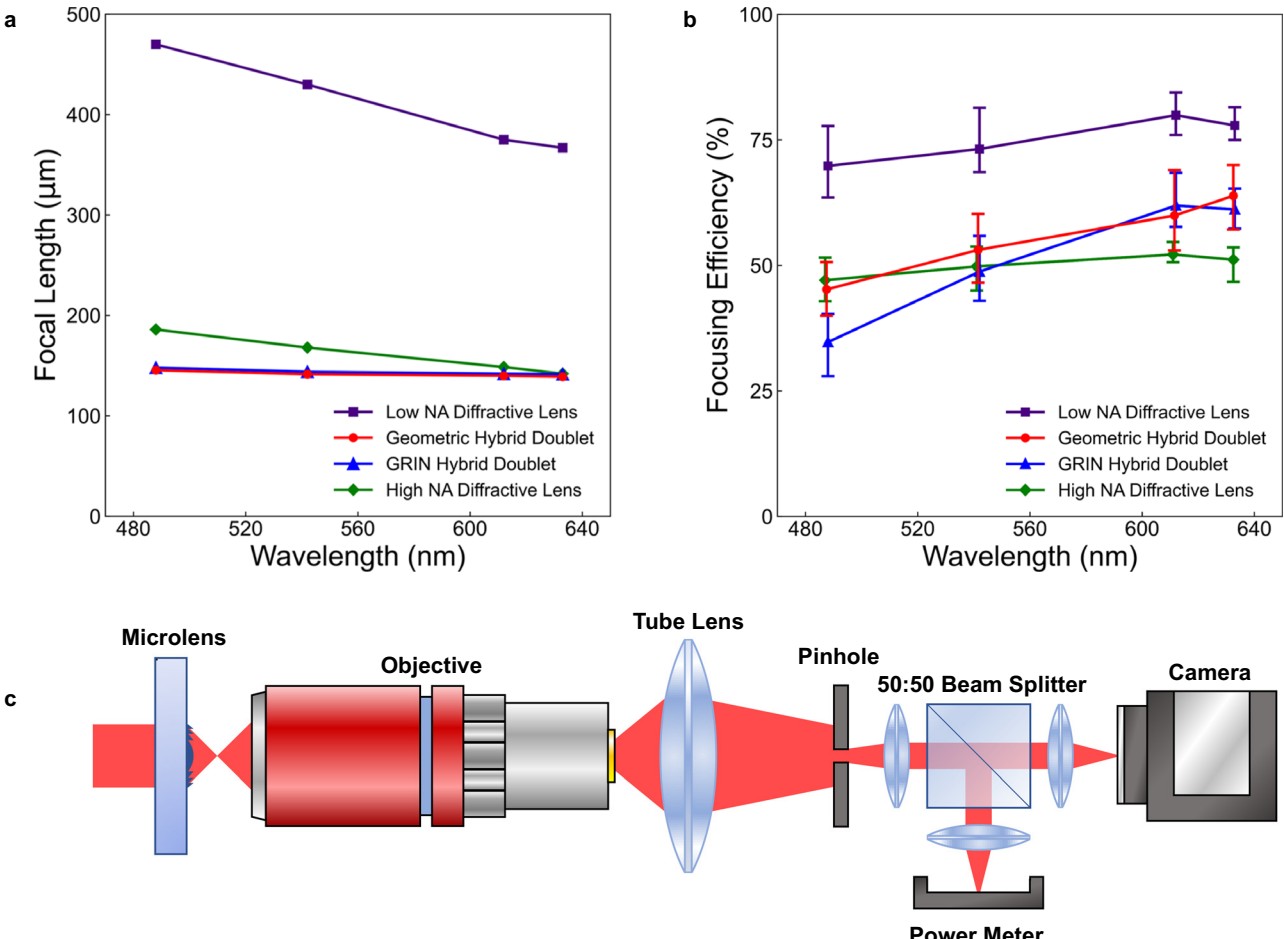

**Fig. 3 | Characterization of focal length and focusing efficiency. a** Graph of focal length plotted against wavelength for lower and higher numerical aperture (NA) diffractive lenses, geometric hybrid lenses, and GRIN hybrid lenses. Each data point was averaged across ten printed lenses. The diffractive lenses experienced stronger negative dispersion than the hybrid lenses. **b** Graph of measured focusing efficiency plotted against wavelength for each type of lens. Each data point was averaged across the ten printed lenses. Error bars represent the maximum and minimum values across the ten samples. **c** Schematic depicting the experimental setup for measuring microlens focusing efficiency. Microlenses (left of the figure) are placed on a motorized stage and illuminated by collimated lasers. A 50:50 beam splitter transmits half of the light to a camera and half to a power meter, allowing the simultaneous alignment of the pinhole and the power measurement.

focusing efficiencies 40–68%) at the measured wavelengths, respectively. Although the GRIN hybrid lenses achieve higher theoretical focusing efficiencies according to our simulations, the measured efficiencies suffered due to additional errors in defining the gradient index profile (see Methods Section). The hybrid lenses also outperformed the control high NA diffractive lenses (green diamonds). The experimentally attainable focusing efficiencies may be further improved as the accuracy of 3D lithography platforms are improved and smoother features can be fabricated. The lens' measured focal spots were also used to estimate the Strehl ratios, available in SI Section 4.

Focusing efficiencies were measured using the custom-built setup for which a schematic is shown in Fig. 3c, which was designed to accommodate the diameters of the printed microlenses. The intensity of light within each lens' focal spot (3 times the FWHM in radius) was divided by the total intensity passing through their apertures to obtain the focusing efficiency. The light outside the measurement region was filtered using a pinhole. A 50:50 beam splitter was used to evenly divide the filtered light between a camera and a power meter, enabling simultaneous alignment and power measurements. Details are available in the Methods Section.

**Broadband visible wavelength imaging**

A 1951 United States Air Force (USAF) resolution target was imaged by the hybrid doublets to characterize their achromatic focusing ability under broadband white light illumination. To accomplish this, both the microlenses and the target were placed on X-Y translation sample holders, allowing the lenses to be aligned to the desired resolution pattern (group 7 element 1, 128 lines/mm). The USAF resolution target was translated along the Z direction using a translation stage to adjust the focus until the fabricated lenses and the USAF target were separated by just over one focal length, creating a real image. An infinity-corrected objective was focused on the image plane of the microlens, magnifying and projecting the image onto a CCD (SI Section 7).

Images formed by the hybrid doublets are compared to the high NA diffractive lens to assess their achromaticity (Fig. 4). Due to chromatic dispersion, the control diffractive lens separated the white light into its color components, creating a chromatically blurred image (illustration in Fig. 4a). A series of six uniquely colored photographs of group 7 were captured by measuring the color-blurred image at different planes (insets in Fig. 4a). In contrast, the hybrid achromats focused all wavelengths nearly to a single point, forming sharper white images under broad illumination (illustration in Fig. 4b). Images formed by the geometric and GRIN hybrid lenses are shown as insets of Fig. 4b. The white light images formed by the doublets are largely free of chromatic aberrations (see SI section 7), demonstrating the power of this approach for forming compact integrated imaging systems.

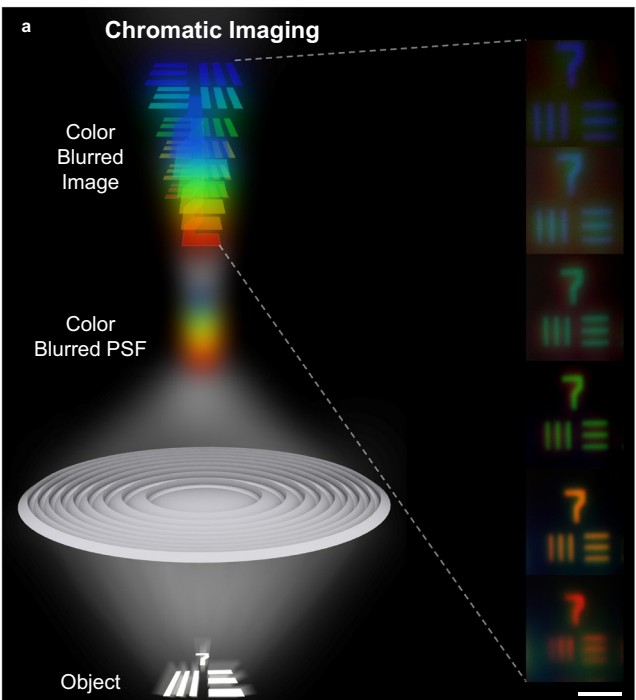

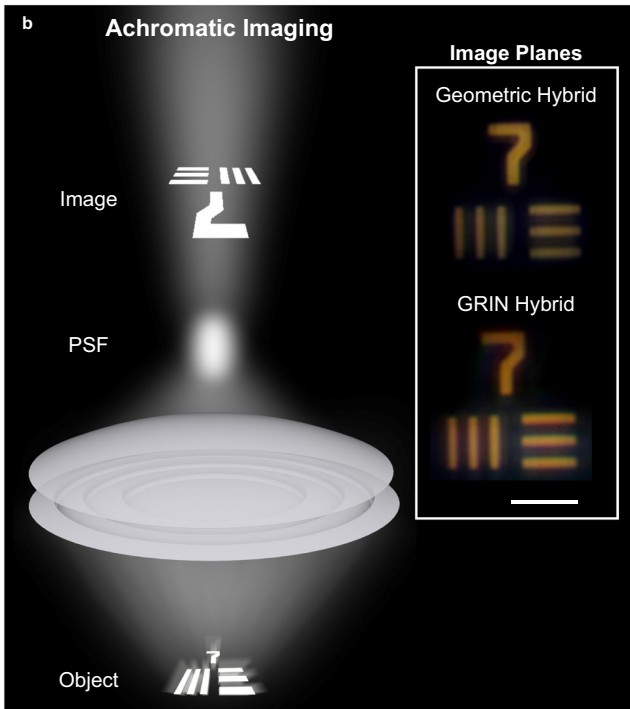

**Fig. 4 | White light imaging with subsurface microlenses. a** Illustration of chromatic imaging by a high NA diffractive lens under broadband white light illumination. The lens focuses light from an object and forms a point spread function (PSF) and an image some distance away. Due to dispersion, the white light is separated into individual color components, resulting in a rainbow color-blurred image. The insets show experimentally collected images measured at different planes 30 μm apart. The chromatic blurring causes a different colored image (red, orange, green, cyan, or blue) to be formed at each plane (no single white light image is formed). The scale bar is 45 μm for the insets. **b** Illustration of achromatic imaging by a hybrid doublet under broadband white light illumination. The PSF and a white image of the object is formed at single planes. Insets show experimentally measured white light images formed by the geometric hybrid achromat and the GRIN hybrid achromat at their image planes. The images have a yellowish tint because a warm white LED was used as the illumination. The scale bar is 45 μm for the insets.

## Individually tuning diffractive and refractive components for increased NA

Hybrid microlenses formed through the stacking of multiple distinct elements can achieve higher NAs for a given aperture size without significant reductions in focusing efficiency or achromaticity, as has been the case for achromatic metalenses[2–4]. As a demonstration, hybrid doublets were fabricated where the geometries of the refractive and diffractive components were tuned to define the NA anywhere between 0.315 and 0.47 at 633 nm (Fig. 5a–h, the hybrid doublet in Fig. 5a is the same as in Fig. 1b). These doublets maintained the same ~15 μm thicknesses but experienced reduced focusing efficiencies within the range of 30–50% (see SI Section 8). The measured focal length error generally increased (and hence image quality generally decreases) with NA due to shifts in the focusing power distribution given by Eq. 1 (more info in SI Section 8). Nonetheless, a focal length error of 7.5% was measured at an NA of 0.471, which is an improvement over the diffractive singlet.

The NA of the hybrid system was tuned by increasing the number of zones in the diffractive element (thereby increasing the focusing power according to Eq. 3) and introducing curvature at the bottom surface of the refractive component (increasing the focusing power according to Eq. 2). The largest drops in focusing efficiency occur as the number of zones in the diffractive lens is increased due to the challenge of accurately printing finer features. Eventually, the spacing between the outer zones would drop below the voxel resolution of the printer as more zones are added. Therefore, tuning the refractive component represents a critical knob for increasing the NA while maintaining good performance with this materials system and fabrication approach. Ultimately, the NA of the two-component imaging system was increased to above 0.47 by using a biconvex refractive component stacked above a diffractive lens with 9 zones (Fig. 5h). The

NA was not increased further in order to maintain high focusing efficiency and achromaticity. Even at this NA, the doublet was able to form a white light image with reduced chromatic errors compared to standard diffractive lenses and exhibited visible wavelength focusing efficiencies between 30 and 40%. The white light images are shown to the right of the confocal fluorescence microscopy images for each lens in Fig. 5a–h.

## Imaging with an achromatic microlens array

Microlens arrays (MLAs) free of chromatic aberrations are essential for high-resolution full-color light-field imaging[47, 48]. However, the fabrication of achromatic polymer MLAs has proven challenging using traditional approaches[49]. As a demonstration, we printed a hexagonal array of 18 subsurface achromatic hybrid microlenses to image the USAF resolution target under white light illumination. The size of the array was selected to cover groups 6 and 7 on the resolution target. An optical microscope image of the fabricated lens array is shown in Fig. 6a.

The USAF resolution target was aligned to the achromatic MLA, producing an array of white light images by collecting light-field information from different regions of the chrome patterns. The patterns of groups 6 and 7 are recreated by the hybrid achromatic MLA. Fig. 6b shows the captured image array and a full image of groups 6 and 7 reconstructed from the array. Different levels of detail and light-field information can also be captured by controlling the distance between the microlens array and resolution target, as shown in SI Section 9. The ability to generate an achromatic array of microlenses that forms images under broadband visible wavelength illumination has powerful implications for applications like light-field cameras and light-field displays. Although we have shown achromatic microlenses with high NAs, large diameters, and high resolution, the design of

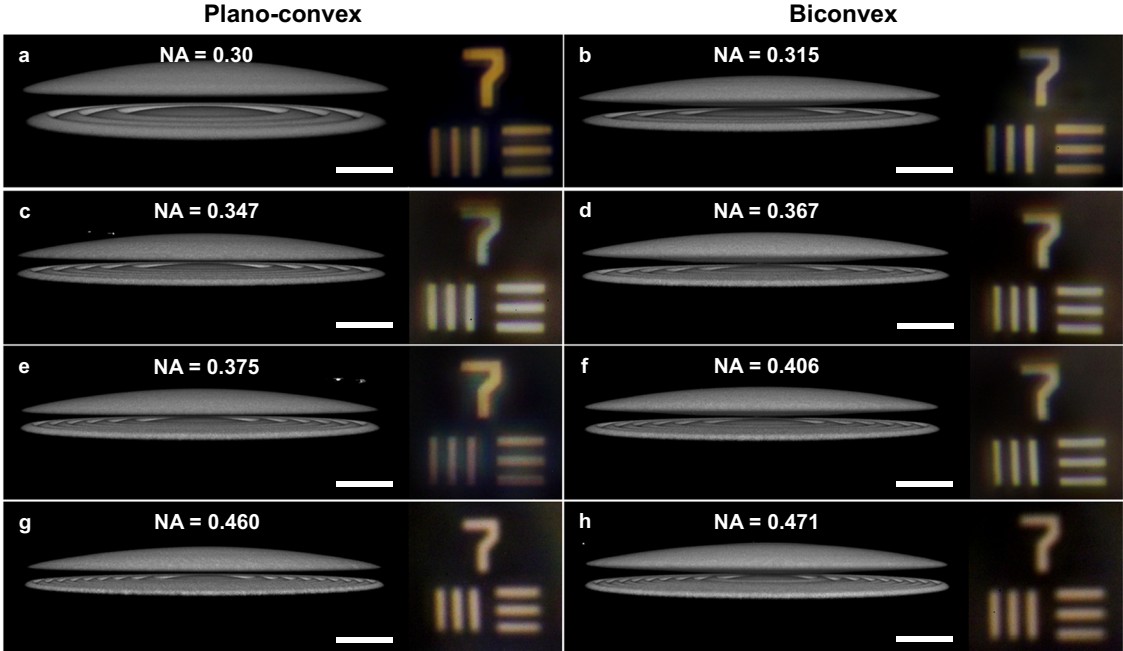

**Fig. 5 | Hybrid doublets with increased numerical apertures (NAs).** Confocal fluorescence intensity images of hybrid doublets with increasingly high NAs from 0.3 to 0.471. Diffractive components were designed with 3 (**a**, **b**), 5 (**c**, **d**), 6 (**e**, **f**), or 9 (**g**, **h**) zones. Refractive components were defined to be either plano-convex (**a**, **c**, **e**, **g**) or biconvex (**b**, **d**, **f**, **h**). Broadband visible wavelength images formed are shown to the right of each respective doublet. Scale bars are 15 μm for the fluorescence images and 45 μm for the images of the resolution target formed by the hybrid microlenses.

lower NA, smaller diameter designs with tighter lens spacings may be easily adopted to acquire more light-field data. Furthermore, the future design of hybrid lenses with extended depth of focus or multiple focal spots may enable light-field cameras that can capture high-quality images of both close and distant objects[50].

## Discussion

As we have demonstrated, the volumetric integration of arbitrary structures in a mesoporous host medium via DLW enables high-performance hybrid achromats that operate at visible wavelengths over a range of at least 488–633 nm. These compact doublets achieved higher NAs (0.3) and higher focusing efficiencies (maximum 51–70%) than achromatic metalenses with equivalent or smaller diameters in the literature (see SI Section 10). The refractive and diffractive components that comprise these hybrid imagers are seamlessly integrated and aligned without requiring external support structures, resulting in a much more compact configuration than is achievable by traditional glass and polymer achromats, including those formed via standard DLW. Focal length errors of less than 3% were measured across visible wavelengths, enabling the lenses to form images under broadband illumination. Elements within the stacked hybrid system can also be individually tuned to alter the overall focusing properties without increasing the total thickness or decreasing the diameter. As an initial demonstration of this capability, a biconvex refractive lens was aligned with a nine-zone diffractive lens to create an achromatic imager with an increased NA of 0.471, but with reduced broadband focusing efficiencies of 30–40% and an increased focal length error of 7.5%. Furthermore, achromatic hybrid microlenses were easily arranged into tightly spaced arrays and reconstructed images were formed, paving the way toward full-color light-field systems.

The presented approach lays the foundation for building more complex, higher-performance microimaging systems. For example, additional refractive elements could be incorporated to increase the NA and provide more surfaces for controlling third-order and chromatic aberrations. The inclusion of inverse-designed GRIN components may also permit hybrid achromats to be scaled up to mm- or cm-

scales without increasing thickness, or to achieve greatly increased NAs on the order of 0.7 or higher, a method that is currently under investigation[51]. Our subsurface lenses can also be easily interfaced with other materials and devices, such as augmented reality headsets. Finally, combining the approach here with substrates containing pre-formed meta-optics could lead to higher NAs, higher focusing efficiencies, better imaging quality, and other functionalities, including polarization control[52].

The multiphoton writing tool used in this study is not originally designed to create structures in porous hosts, and variations in the femtosecond-pulsed laser writing process between prints make achieving the designed refractive index challenging. There are also variations within a single porous silicon oxide films and batch-to-batch inconsistencies in porous silicon oxide fabrication[46]. New calibration methods are being developed that significantly reduce the variation in the printed refractive index both within a sample and across batches of samples[53]. Thus, addressing these challenges should result in the measured focusing efficiencies approaching the simulated values of nearly 90% across the visible. Advancements in the multiphoton laser technology to improve fabrication area and throughput could enable the production of larger area lenses and achromatic MLAs[54–56]. We believe the achromats presented in this work will inspire further research into hybrid diffractive/refractive systems that manipulate light for applications, including near-eye displays, lightweight cameras, light-field imagers, and light-field displays.

## Methods
### Microlens fabrication
Porous silicon (PSi) films of controlled thicknesses were formed by electrochemically etching bulk Si (boron doped, 0.001 Ω cm resistivity, (100)-orientation, 500 μm thick) under ethanolic hydrofluoric acid (60% aqueous hydrofluoric acid, 40% ethanol) using a current density of 400 mA cm⁻². The films were detached from the substrate through an electropolishing electrolyte (15% aqueous hydrofluoric acid, 85% ethanol, current density of 20 mA cm⁻²) and transferred onto transparent fused silica substrates via a gentle stream of ethanol. The

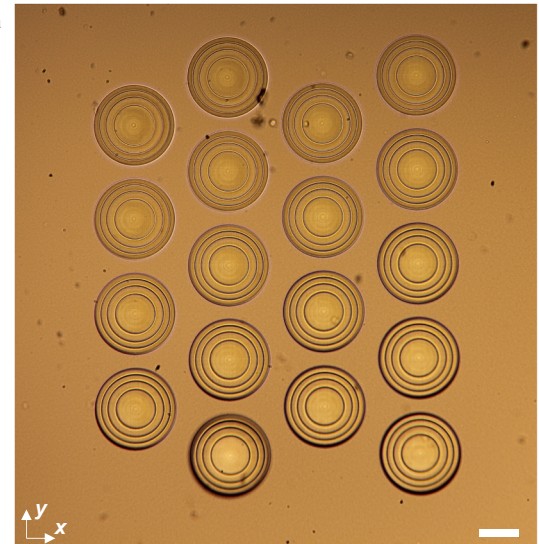

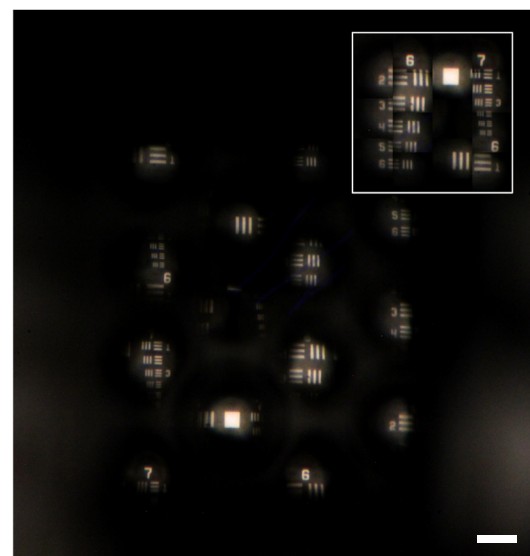

**Fig. 6 | Imaging with an achromatic microlens array. a** Optical microscope image of the achromatic microlens array illuminated by a white LED. Each microlens in the array is identical to the lens in Fig. 1b. **b** Array of images formed by the microlens array in **a** under broadband white light illumination. Light-field information of groups 6 and 7 on the USAF resolution target is collected. The inset shows a rendered image that was reconstructed from the collected array of images. Scale bars are 45 μm.

transferred films were carefully dried to prevent pore collapse from capillary forces. The PSi films attached strongly to their new substrates upon drying. PSi was converted to $PSiO_2$ by thermal oxidation at 900 °C for 30 min under oxygen. IP-Dip photoresin was dripped onto the prepared $PSiO_2$ host mediums and allowed to soak for 30 min, allowing the photoresin to completely fill the pores. A Nanoscribe Photonic Professional GT was used in the dip-in configuration to 3D print the lenses inside the pores of the host films. A 63x objective focused a 780 nm wavelength femtosecond titanium-sapphire laser into the volume of the $PSiO_2$. Galvanometric mirrors scanned the laser to selectively polymerize the photoresin inside the pores. A constant laser power of 15 mW and a scan speed of 10 mm/s was used to print the diffractive components and the geometric refractive components, while the GRIN refractive lenses were formed by spatially changing the laser power between 10 and 22.5 mW while printing according to the equation:

$$LP(r) = (LP_{max} - LP_{min})\left(1 - \frac{r^2}{R^2}\right) + LP_{min} \qquad (7)$$

where LP is the laser power, $LP_{max}$ is 22.5 mW, $LP_{min}$ is 10 mW, R is the maximum radius, and r is the radius. The lens diameters were restricted to a maximum value of 90 μm to ensure that printed features are centered within the write field of our femtosecond laser printer, eliminating stitching errors[57] and minimizing laser dosage inconsistencies. For the GRIN lens, the slice distance in Z was 1 μm, while the slice distance was 100 nm for all other optics. The diffractive and refractive components were naturally aligned within the volume of the host films during fabrication, creating high NA hybrid lenses without the need for additionally fabricated support structures. Each hybrid lens took around 10 min to print, with the array in Fig. 6 taking ~3 h. After printing, films were developed in propylene glycol monomethyl ether acetate at 80 °C for 2–3 h, allowing unpolymerized resist to diffuse out of the pores. Developed films were solvent exchanged with isopropanol, and supercritical dried to avoid cracking around the printed lenses.

### Focusing efficiency
Printed microlenses were illuminated by free-space lasers with wavelengths of 633 nm (He-Ne), 612 nm (He-Ne), 542 nm (He-Ne), and 488 nm (Argon). The intensity of the lasers was filtered on the illumination side such that the CCD on the imaging side was not saturated. To measure the power within the aperture of a printed microlens, an objective (either 50x, 0.42 NA, infinity-corrected Mitutoyo or 50x, 0.60 NA, infinity-corrected Olympus) was first focused onto the sample plane, after which the magnified image was focused by a tube lens. An iris located at the image plane of the tube lens was opened to the same size as the magnified microlens. An achromatic lens positioned one focal length away from the iris recollimated the light. A 50:50 beam splitter divided the light evenly between the CCD and a power meter, allowing for simultaneous alignment and power measurement. Identical lenses on each side of the beam splitter focused the light onto the power meter and the CCD, each of which was placed at the image plane of their respective lenses. Once the iris was opened to the appropriate size, the fabricated microlens was moved away, and the objective was focused onto a blank area of the substrate near the sample and the background power was recorded. The objective was focused back onto the microlens sample, and a motorized stage translated the sample until the objective was focused onto the focal spot of the microlens. The iris was replaced by a pinhole with a radius three times the FWHM of the magnified focal spot and the power was recorded. The intensity of light within each lens' filtered focal spot was divided by the total intensity passing through their apertures to obtain the focusing efficiency.

### White light imaging
The printed microlenses were illuminated by a collimated warm white light LED (Thorlabs, 400–700 nm). The white light was focused onto a negative USAF target to increase the intensity of the illumination. The resolution target and the substrate containing the fabricated microlenses were each mounted onto separate X-Y translation holders, allowing the lenses to be aligned to group 7 element 1. The translation holder supporting the target was placed on a translation stage, enabling control over the distance between the target and the microlens. The target was adjusted to roughly one focal length away from the microlens substrate. An objective magnified the image of the USAF target formed by the fabricated microlens and projected the image onto a camera's CCD. The brightness and contrast of the collected images were adjusted by controlling the shutter speed and ISO of the camera and by adjusting the brightness of the LED.

## Confocal imaging

X-Z focal profiles and X-Y focal spot images were captured using a WITec Alpha 300 S upright confocal microscope. The lenses were illuminated by the same four free-space gas lasers, and the light was collected by the 60x, 0.6 NA Olympus objective. A multimode collection fiber with a 25 µm core diameter acted as a pinhole for the confocal imaging. The collected light was transmitted to an avalanche photodiode. A piezo stage was used to translate the sample in X and Y during imaging, allowing the focal spots of the lenses to be captured. For the captured X-Z focal profiles of the lenses, a stepper motor moved the sample stage in Z after each line was captured, forming a Z stack of images. Images were falsely colored according to the measurement wavelength because the measurements only contain intensity data, not color data (e.g., focal profiles measured at 612 nm were edited to have an orange coloring).

## Focal length measurements

Printed microlenses were illuminated by free-space lasers with wavelengths of 633 nm (He-Ne), 612 nm (He-Ne), 542 (He-Ne), and 488 nm (Argon). A WITec Alpha 300 S upright confocal microscope was focused onto the sample plane and the Z-position was recorded. The confocal microscope was then focused near the focal plane of the microlenses and the intensity profile was collected. The Z-position was adjusted slightly until the intensity was maximized and the Z-position was recorded. The focal length was taken as the measured difference between the sample plane and the focal plane.

## Multiphoton confocal fluorescence imaging

A Zeiss LSM 710 NLO inverted confocal microscope was used for multiphoton fluorescence imaging. An ultrafast titanium-sapphire laser (Newport, Mai Tai) with 780 nm excitation wavelength was focused by an objective (63x, 1.4 NA oil immersion) onto the printed lenses. A low average laser power of 2–10% was used to avoid damaging the microlenses. Z-stacks were formed by translating the sample in X, Y, and Z during imaging. Images were processed and rendered using Amira software.

## Strehl ratio calculations

The experimentally measured field intensity in the focal plane was smoothed to reduce the effect of single-pixel noise in the measurements. Smoothing was performed using a simple cone filter with a radius of five data points. To disregard the noise floor in the data, a constant shift was applied to the focal spot by subtracting its minimum value. The post-processed data was centered at the main focal spot and the data was truncated using a fixed radial distance from the center of the focal spot. The total intensity of the measured data was then computed by numerical integration. After normalizing the intensity, the peak intensity of the Airy disk (the ideal focal pattern) corresponding to the measured wavelength and NA was computed. Finally, the Strehl ratio was estimated by dividing the maximum intensity of the processed measured data by the maximum intensity of the ideal Airy disk, with identical total field intensity.

## Image reconstruction

A white LED was used to illuminate a USAF resolution target. A microlens array simultaneously captured images of different regions of the target. Square patches of equivalent size surrounding the centers of each microlens were stitched together. Each sub-image was then rotated 180°, and finally, the full reconstructed image was rotated 180° to assume the correct upright orientation.

## Simulations

Focusing efficiency simulations were performed by building the 3D lens models in the software Ansys Lumerical and using 3D FDTD analysis. An auto-generated mesh of at least ten points per wavelength was used throughout the device. The electromagnetic field was captured just beyond the thickness of each simulated lens and then projected into the far field using a radiation integral approach[58]. The size of the 3D simulations necessitated running them on cloud compute instances distributed on 120 ARM-based CPU cores and using 480 GB RAM. Broadband simulations were used, except for in the case of the GRIN lens, where spatially variant indices were generated for single-frequency FDTD simulations. For the doublet simulations, typical wall time was between 1–2 h, but significant variability was observed. At 488 nm, the GRIN doublet simulation was computationally intensive and the high-performance computing instances ran out of memory. For the singlets, simulation times ranged between 5–30 min. The fields were projected from the nearfield onto a 200 × 200 µm square in the focal plane using a higher resolution 12 × 12 µm central region surrounding the focal spot. Surrounding the focal spot, a resolution of 200 nm was used, with 400 nm spacing outside, resulting in 512 × 512 spatial points at each wavelength. From here, the power passing through an aperture with three times the FWHM in radius was calculated and divided by the total power passing through the entire 200 µm screen. Given the rapid fall of the main lobe with limited resolution, the FWHM could only be measured with a precision of 200 nm, which could have contributed to errors in the calculations. The interface between the host medium and air was ignored to reduce the simulation size and complexity, leading to a variation between simulation and measurement in the focal length. In addition, COMSOL Multiphysics was used to simulate the focal profiles of the lenses.

## Data availability

The data that support the findings of this study are available from the corresponding author upon request.

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

## Acknowledgements

Preliminary work supported by a University of Illinois Urbana-Champaign Grainger College of Engineering Strategic Research Initiative (L.L.G. and P.V.B.). Optical modeling supported by the National Science Foundation (No. ECCS-1935289) (L.L.G. and P.V.B.). Lens fabrication and characterization supported by the "Photonics at Thermodynamic Limits" Energy Frontier Research Center supported by the US Department of Energy, Office of Science, Office of Basic Energy Sciences under Award Number DE-SC0019140 (P.V.B.). C.A.R. acknowledges the support of the National Defense Science and Engineering Graduate (NDSEG) Fellowship Program through the United States Department of Defense.

## Author contributions

C.A.R. and C.R.O. conceived the design strategy for forming achromatic hybrid doublet microlenses. C.A.R. and D.X. fabricated and characterized microlens samples. C.A.R., C.R.O., and D.G.C. designed

the focusing efficiency measurement setup. C.A.R. and R.E.C. measured and calculated the Strehl ratios for microlenses. C.A.R. and T.R. performed the simulations. H.G. assisted with the fabrication, preparation, and characterization of microlenses. C.A.R. and C.R.O. wrote the manuscript. L.L.G., R.E.C., D.G.C., and P.V.B. provided critical feedback to improve the project. P.V.B. supervised the project.

## Competing interests

L.L.G., P.V.B., C.H.O., and C.A.R. claim a US patent on some of the processes and devices presented in this work through the University of Illinois Urbana-Champaign. The remaining authors declare no competing interests.
