## [Peer Review File · Nature Communications]

Hybrid Achromatic Microlenses with High Numerical Apertures and Focusing Efficiencies Across the VisibleEditorial Note: This manuscript has been previously reviewed at another journal that is not operating a transparent peer review scheme. This document only contains reviewer comments and rebuttal letters for versions considered at *Nature Communications*.

Reviewer #1 (Remarks to the Author):

I was Reviewer #1 from a previous submission. I believe that the authors have satisfactorily addressed all points from the previous review and that the manuscript rises to the level of Nature Communications.

Reviewer #2 (Remarks to the Author):

The revised paper and the response letter reflected all the comments from the reviewers. The following improvements could make the paper worthy of publication in Nature Communications:

- 1) As reviewer 1 said, the advantage of the technique in this paper over the papers cited is not clear. Please compare the characteristics of the lenses using the technology described in this paper with the cited results for NA, diameter, chromatic focal shift, and focusing efficiency in a table or plot them in a graph so that the reader can clearly see the advantage of this technology. In particular, a large lens has been shown in "Fig. 5 1-cm-diameter RGB-achromatic polarization-insens (NA= 0.3)" in Reference 15, which seems to be a good comparison for the lens in this paper.**
- 2) My understanding is that the advantage of the proposed technology is that the optical system can be realized in a thin combination lens configuration to achieve function and performance, and that the lenses can be positioned with high precision. If so, the authors should emphasize the advantages of combination lenses, for example, in the abstract or at the beginning of the paper to make this point clear. It would also be better if the title of the abstract clearly states that it is a combination lens configuration and that it was fabricated using a 3D printer.**

We thank the reviewers for their insightful responses to our work and for their helpful suggestions on how to improve the quality of the manuscript. We have revised the content in response to the reviewers' feedback as detailed below. Reviewers' comments in italics followed by our responses in normal font.

Reviewer #1:

I was Reviewer #1 from a previous submission. I believe that the authors have satisfactorily addressed all points from the previous review and that the manuscript rises to the level of Nature Communications.

Reviewer #2:

The revised paper and the response letter reflected all the comments from the reviewers. The following improvements could make the paper worthy of publication in Nature Communications:

1) As reviewer 1 said, the advantage of the technique in this paper over the papers cited is not clear. Please compare the characteristics of the lenses using the technology described in this paper with the cited results for NA, diameter, chromatic focal shift, and focusing efficiency in a table or plot them in a graph so that the reader can clearly see the advantage of this technology. In particular, a large lens has been shown in "Fig. 5 1-cm-diameter RGB-achromatic polarization-insens (NA= 0.3)" in Reference 15, which seems to be a good comparison for the lens in this paper.

Response: We have added a table in Supplementary Information Section 10 that compares the lenses explored in this work with examples from the literature in terms of wavelength range, diameter, numerical aperture, focusing efficiency, and achromaticity.

2) My understanding is that the advantage of the proposed technology is that the optical system can be realized in a thin combination lens configuration to achieve function and performance, and that the lenses can be positioned with high precision. If so, the authors should emphasize the advantages of combination lenses, for example, in the abstract or at the beginning of the paper to make this point clear. It would also be better if the title of the abstract clearly states that it is a combination lens configuration and that it was fabricated using a 3D printer.

Response: We have revised the manuscript in multiple locations to further clarify these points. In the abstract, we have included the sentences, "Here, subsurface 3D printing inside mesoporous hosts is used to densely integrate aligned refractive and diffractive elements, forming thin high performance hybrid achromatic imaging micro-optics." and "The presented approach precisely combines optical components within 3D space to achieve thin lens systems with high focusing efficiencies, high numerical apertures, and low chromatic focusing errors, providing a pathway towards achromatic micro-optical systems."

At the end of the introduction we also say, "This study presents a path to forming compact imagers that overcomes many limitations of metalenses and traditional polymer micro-optics by using 3D printing to combine multiple optical elements within the pore volume of a host medium, achieving high, controllable NAs, high focusing efficiencies, good dispersion control, and large diameters while retaining a thickness of only 15 μm ."